# Use of a Silver-Impregnated Vascular Graft: Single-Center Experience

**DOI:** 10.3390/antibiotics11030386

**Published:** 2022-03-15

**Authors:** Jiri Molacek, Vladislav Treska, Karel Houdek, Václav Opatrný, Bohuslav Certik, Jan Baxa

**Affiliations:** 1Vascular Surgery Department, University Hospital in Pilsen, Faculty of Medicine in Pilsen, Charles University, Husova 3, 301 00 Plzeň, Czech Republic; treska@fnplzen.cz (V.T.); houdekk@fnplzen.cz (K.H.); houdek@fnplzen.cz (V.O.); certik@fnplzen.cz (B.C.); 2Department of Imagine Methods, University Hospital in Pilsen, Faculty of Medicine in Pilsen, Charles University, Husova 3, 301 00 Plzeň, Czech Republic; baxaj@fnplzen.cz

**Keywords:** antibiotics, graft patency, silver-impregnated vascular graft, vascular graft infection

## Abstract

Introduction: Vascular graft infection is a life threatening situation with significant morbidity and mortality. Bacterial graft infection can lead to false aneurysms, bleeding and sepsis. There are a lot of risky situations where grafts can become infected. It is therefore highly desirable to have a vascular graft that is resistant to infection. In this retrospective clinical study, a silver-impregnated vascular graft was evaluated in various indications. Methods: Our study included a total of 71 patients who received a silver-impregnated vascular graft during the period from 2013 to 2018. Patients had an aortoiliac localization of vascular graft in 61 cases (86%), and a peripheral localization on the lower limbs in 10 cases (14%). Indications for the use of these special vascular grafts were trophic lesions or gangrene in the lower limbs in 24 cases (34%), suspicious mycotic abdominal aortic aneurysm (mAAA) in 4 cases (5.5%), salmonela aortitis or aneurysms in 4 cases (5.5%), infection of the previous vascular graft in 11 cases (15.5%), other infections in 12 cases (17%), AAA rupture in 10 cases (14%) and other reasons (pre-transplant condition, multiple trauma, graft-enteric fistula) in 6 cases (8.5%). Thirty-day mortality, morbidity, the need for reintervention and amputation, primary and secondary graft patency, and finally the presence of a proven vascular graft infection were evaluated. Results: The 30-day mortality was 19.7%, and morbidity was 42.2%. The primary patency of the vascular graft was 91.5%. Reoperation was necessary in 10 cases (14%) and amputation was necessary in 10 cases (14%). The median length of hospital stay was 13 days and the mean follow-up period was 48 ± 9 months. During the follow-up period, six patients (8.5%) died from reasons unrelated to surgery or without any relation to the vascular graft. Secondary patency after one year was 88%. Infection of the silver graft was observed in three patients (4.2%). Conclusions: Based on our results, the silver graft is a very suitable alternative for solving infectious, or potentially infectious, situations in vascular surgery. In particular, in urgent or acute cases, a silver graft is often the only option.

## 1. Introduction

Despite ever-improving vascular grafts, better antibiotic prophylaxis and better and earlier specific diagnostic methods, prosthetic vascular graft infection remains one of the worst surgery procedure-related complications. A way to prevent and possibly address these serious situations is still being discussed. The use of venous autografts is obviously limited, and the use of allografts carries its own specific problems and complications [1,2,3,4,5,6].

Creating an artificial vascular graft that is resistant to infection is one of the goals of current research in the field of vascular surgery. Infection in an artificial vascular graft has significant mortality and morbidity [7]. Bacterial graft infection can lead to false aneurysms, bleeding, chronic infected wounds and sepsis. These are situations that directly threaten the patient’s life, and infection of the vascular reconstruction in the lower limbs is often associated with the risk of limb loss. Despite obvious antibiotic prophylaxis, we observe this complication in 1.5–6% of patients [3,7,8]. The most common infectious species are Stafylococcus aureus, methicillin-resistant *Staphylococcus aureus*, *Psedomonas aeruginosa*, *Salmonella enteritidis* and others. Infections caused by fungi are less common [2,9]. According to most authors, the permanent and definitive treatment of an artificial vascular graft infection is impossible [4,10,11]. Therefore, graft explantation and replacement is almost always necessary. However, this is associated with the problem of replacing an infected graft. In the aortoiliac region, venous grafts cannot be used, with a few exceptions [4,5]. Operative techniques using the superficial femoral vein are still uncommon [12]. We are forced to continue using an artificial vascular graft or arterial allograft, which is often not available in urgent or acute cases [13]. It is therefore highly desirable to have a vascular graft that is as resistant to infection as possible. Implantation of these specific grafts has more indications than just the replacement of infected vascular prostheses. The most common are vascular reconstructions in lower limbs with trophic lesions or gangrene, acute aortitis, mycotic aneurysms (Figure 1), aortoduodenal fistula, and others [7,14]. Antibiotic-impregnated vascular grafts were introduced many years ago; this option is still commonly used and we have a lot of literature data with relatively satisfactory results [15,16,17,18,19]. The problem is the relatively short period in which antibiotics are released from the graft. Grafts impregnated with silver salts have been introduced relatively recently. The silver salts, firmly bound in polyester vascular grafts, act directly on the phospholipid layer of the cytoplasmic membrane of bacteria and also directly attack bacterial DNA and prevent its replication. Therefore, this is a completely different mechanism compared to the release of antibiotics from antibiotic-bound grafts [20]. The microbicidal function of silver should therefore significantly increase resistance to infection [15,21], but recently published studies are also controversial [22,23]. The aim of this work is to present the results of an observational retrospective clinical study conducted in a single center using silver-impregnated vascular grafts in various indications. As for us, the novelty of the study is the use of silver-impregnated grafts in various risky indications with very good results. We did not avoid the use of silver grafts, even in severe cases, such as a confirmed mycotic aneurysm, infection of a regular vascular graft with methicillin-resistant *Staphylococcus aureus* or major intra-abdominal trauma requiring vascular reconstruction. The authors also discuss the question of necessary long-lasting antibiotic treatments in these specific situations.

## 2. Methods

This is a monocentric retrospective clinical study that retrospectively analyzed the data of patients in whom a silver-impregnated vascular graft was implanted for various indications.

Our study included a total of 71 patients, 53 men (75%) and 18 women (25%), who received a silver-impregnated vascular graft during the period from 2013 to 2018 at the Vascular Surgery Dpt., University Hospital, in Pilsen.

The mean age of the patients was 65.7 years. Patients had an aortoiliac localization of the vascular graft (aortobifemoral bypass, aortoaortic, aortofemoral, iliacofemoral bypass) in 61 cases (86%), and a peripheral localization in the lower limbs (femoropopliteal, femorofemoral or femorocrural bypass) in 10 cases (14%) (Figure 1). We have summarized crucial data from the cohort in Table 1. Indications for the use of these special vascular grafts were trophic lesions or gangrene in the lower limbs in 24 cases (34%), suspicious mycotic abdominal aortic aneurysm (mAAA) in 4 cases (5.5%) (Figure 2), salmonella aortitis or aneurysms in 4 cases (5.5%), infection of the previous vascular graft in 11 cases (15.5%), other infections in 12 cases (17%), AAA rupture in 10 cases (14%) and other reasons (pre-transplant condition, multiple trauma, graft-enteric fistula) in 6 cases (8.5%) (Figure 3). All surgical procedures were performed under general anesthesia; antibiotic therapy was used in all cases. In 35 cases (49.5%), only prophylactic wide spectrum antibiotics were used, which included two doses perioperatively. In the remaining 36 cases (50.5%), antibiotic therapy continued for longer, at least for one week, due to severe infection in various localizations. Cefazolin was used as a prophylactic antibiotic, while in other cases the antibiotics were used specifically based on previous culture results (cefuroxime, vancomycin, ciprofloxacin and combinations of them).

The reasons for using silver grafts are described in Figure 3.

All data were obtained from the University Hospital clinical database. Thirty-day mortality, morbidity, the need for reintervention and amputation, primary and secondary graft patency, and finally the presence of a proven vascular graft infection were evaluated.

## 3. Results

30-day mortality in our cohort was 19.7% (14 patients), the most common causes of death being hemorrhagic shock, sepsis and multiorgan failure. Our mortality rate was fully comparable with literature data on these major vascular emergencies [4,5,24]. Morbidity was 42.2% and most commonly included sepsis, multiorgan failure, respiratory failure, and wound infection. This data corresponded to similar cohorts of patients (rupture of AAA, mycotic AAA, vascular graft infection) where silver grafts had not been used [25,26]. In the same time period, 186 regular non-silver grafts were implanted in these indications. A summary of the morbidity/mortality results in the silver graft group is presented in Table 2.

The primary patency rate of vascular grafts at 1 year was 91.5%. Early graft closure (within 30 days) occurred in six cases. Predictors of primary patency loss were the presence of critical limb ischemia, diabetes mellitus and the necessity of below-the-knee bypasses. There was no difference in patency between the silver graft and regular graft cohorts. Reoperation was necessary in 10 cases (14%); the reason was mainly graft occlusion (8.5%), bleeding (10%), wound infection (8.5%) or a combination of the above. Amputation was necessary in 10 cases (14%). The median length of hospital stay was 13 days and the mean follow-up period was 48 ± 9 months. During the follow-up period, six patients (8.5%) died from reasons unrelated to surgery or without any relation to the vascular graft. Secondary patency after 1 year was 88%. All of this data corresponded with data for common cohorts, where a normal graft was used (our own unpublished data).

Proven infection of the silver graft was only observed in three patients (4.2%) during the follow-up period (Table 3). The infections were confirmed by positron emission tomography with computed tomography (PET/CT)—high 18F-fluoro-2-deoxy-d-glucose (FDG) uptake was observed in vascular grafts and also by positive cultures. *Staphylococcus aureus* was cultured in two cases (haemoculture) and *Salmonella enteritidis* in one case as a monoinfection not only from blood cultures but also from the silver graft. Thus, there was no doubt that these agents were the cause of infection. In one case, the vascular graft was explanted and replaced with an aortic allograft. Figure 4 shows the PET/CT of the infected silver graft before explantation, where the cultivated bacteria were *Salmonella enteritidis*. The probable source of infection was diarrhea 2 months before clinical signs of sepsis occurred. The other two cases of peripheral silver graft infection were treated conservatively with very good results (vancomycin 0.5 mg/day intravenously for 10 days, then ciprofloxacin 800 mg/day orally for a minimum of 6 weeks). Regular CT and PET/CT examinations were performed to follow the treatment success. All three patients with silver graft infections recovered very well without any severe complications. Patients with an aortic allograft had no clinical or PET/CT signs of re-infection, and both patients with left infected peripheral vascular reconstructions had no clinical signs of infection, although the PET/CT images indicated persistent high uptake values of 18F-fluoro-2-deoxy-glucose (18F-FDG). They are, at present, without antibiotic treatment.

## 4. Discussion

Infection of any artificial materials implanted into the human body is always a very serious problem, whether it is an artificial joint, heart valve, hernia mesh or artificial vascular graft. Especially in vascular surgery, infection of an artificial vascular graft is a nightmare for every vascular surgeon. These situations are associated with significant morbidity and mortality [27]. The use of resistant vascular grafts may be beneficial as it may reduce the need for reoperation caused by vascular graft infection and can improve the outcome of surgeries in potentially infectious areas (trophic lesions of the lower limbs, gangrene, aortitis, aortoduodenal fistula, ongoing bacteremia, etc.).

The use of cryopreserved or fresh allografts is one of the alternatives for managing these catastrophic situations. According to a number of authors, the results are very good and comparable to the use of silver grafts based on the results of a number of studies [28]. The problem is an obvious lack of these grafts in acute and urgent cases. Antibiotic-impregnated grafts have not met expectations in the past, and according to recent data, are inferior to silver-impregnated grafts [1,24]. In in vitro studies, different types of grafts were each contaminated separately with various micro-organisms. For all micro-organisms tested (authors have tested Staphylococcus epidermidis, methicillin-resistant *Staphylococcus aureus* (MRSA), *Escherichia coli*, and *Candida albicans*), the silver graft demonstrated a more sustainable and efficient 7-day antimicrobial activity than the rifampicin-soaked graft [15,29]. This is probably related to the rapid release of antibiotics with early peak values in the blood and a rapid decline of blood levels. There are many in vitro studies showing the bactericidal effect of silver [29,30,31,32].

Our own clinical experiences with antibiotic grafts are also very poor. We only implanted 12 rifampicin-soaked vascular grafts in this time period, but the results were unsatisfactory; six patients showed signs of reinfection within the 1-year follow up.

On the other hand, a number of data are available that demonstrate the very good results of silver grafts, the good long-term patency of these vascular grafts and especially a low percentage of infection [1,7,22]. These data are often based on large multicentric trials or meta-analyses with similar inclusion criteria, so their results are very comparable and their conclusions are credible.

Our retrospective study focused on the long-term results of silver graft use in various indications. The primary aim was to obtain information about the long-term patency of these vascular grafts, other complications associated with the grafts and especially about their infection. Our values of primary and secondary patency (91% and 88%, respectively) are fully in line with the use of common vascular grafts in non-risk situations. We did not observe any allergic reactions to the silver grafts in our cohort. Infection in the graft occurred only in three patients (4.2%), despite very risky infectious situations. In one of them, we had to explant the graft and use a fresh aortic allograft, while in the remaining two patients conservative therapy was used (long-term antibiotic treatment).

Our very good experience with silver grafts makes us consider the possible expansion of indications for the use of these highly-resistant vascular grafts; for example, in immunosuppressed patients or in combination procedures, where the risk of infection is multiplied (AAA resection plus another procedure e.g., cholecystectomy, colectomy, liver resection, etc.). The novelty of the study is the use of silver-impregnated grafts in various risky indications with very good results.

The immediate availability of silver grafts in their various variants is of particular importance, especially in the field of urgent and acute vascular surgery. It is known that the floridity of the inflammatory process in the vascular graft area can fluctuate, and thus in some cases, the situation can be managed with long-term antibiotic therapy. However, there are situations where a sudden onset of an attenuated infection does not give time to plan an elective solution and obtain an allograft for urgent surgery, especially in the case of sudden bleeding, e.g., from a rupture of a pseudoaneurysm in one of the anastomoses. A similar situation can occur in the case of the development of prosthetic-enteric fistula with associated, mostly massive, bleeding into the affected part of the gastrointestinal tract. Equally urgent are cases of AAA rupture, where we perioperatively find the cause of the aortitis, most often of salmonella origins. Similarly, a previously unknown infection in a stent-graft implanted in AAA can only manifest by its rupture, especially in non-compliant patients who have not been monitored on a regular basis. Infection in stent-grafts is often mentioned as a severe EVAR complication [33]. To avoid it, strict aseptic conditions during stent-graft delivery are necessary.

However, silver grafts are not only used in aortoiliac surgery, but also in peripheral blood vessels. It is well-known that many patients do not have a venous graft available for vascular reconstruction, whether that is due to gracility, varicosity or saphenous vein trunk stripping. Here again, the use of silver grafts is available as a suitable alternative, especially in patients with trophic lesions in the lower limbs, whether of ischemic, diabetic, venous or combined etiology.

The use of silver grafts is contraindicated in known allergies to silver and palladium, which is very rare in the population. We have not observed this situation. The disadvantage is the slightly higher price of silver grafts compared to a conventional vascular grafts, but when compared with the costs associated with obtaining an allograft or the treatment of infectious complications of common vascular grafts, the price of silver grafts is still relatively lower [28]. More recently, triclosan (5-chloro-2-(2,4-dichlorophenoxy)phenol) has been combined together with silver in a new generation of grafts with promising results [34].

## 5. Conclusions

Based on our results, silver grafts are a very suitable alternative for solving infectious or potentially infectious situations in vascular surgery. High primary and secondary patency, low graft-infection rate and optimal costs are the crucial factors that allow us to draw such conclusions. Especially in urgent or acute cases, silver grafts are often the only option. This is, in our opinion, the biggest advantage of using silver-impregnated grafts. Even if it was just a bridge to buy time for a possible final solution, it is worth having this method in our portfolio. Furthermore, the physical condition of the patient must be considered in the decision of which strategy will be selected. Our study shows very reasonable and acceptable results in situations where infection would certainly occur using a common vascular graft. A large randomized trial in this field would certainly be desirable; the problem is if this study is feasible with the heterogeneity of the group of patients in whom silver graft use is indicated. Therefore, a number of biases can logically be expected. Based on our own experience, and also on the basis of a number of literary data, the use of silver-impregnated grafts is beneficial. Future results can be even better thanks to a new generation of vascular grafts where silver salts are combined with other bactericidal substances.

## Figures and Tables

**Figure 1 antibiotics-11-00386-f001:**
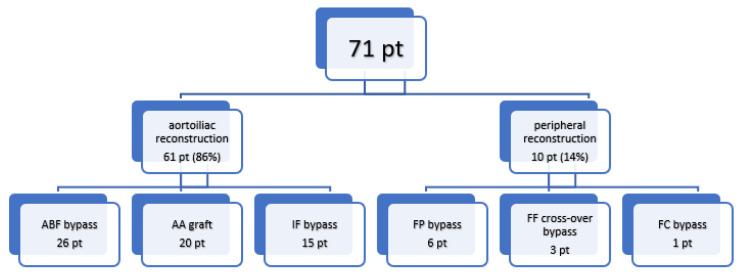
Types of reconstructions included in the study.

**Figure 2 antibiotics-11-00386-f002:**
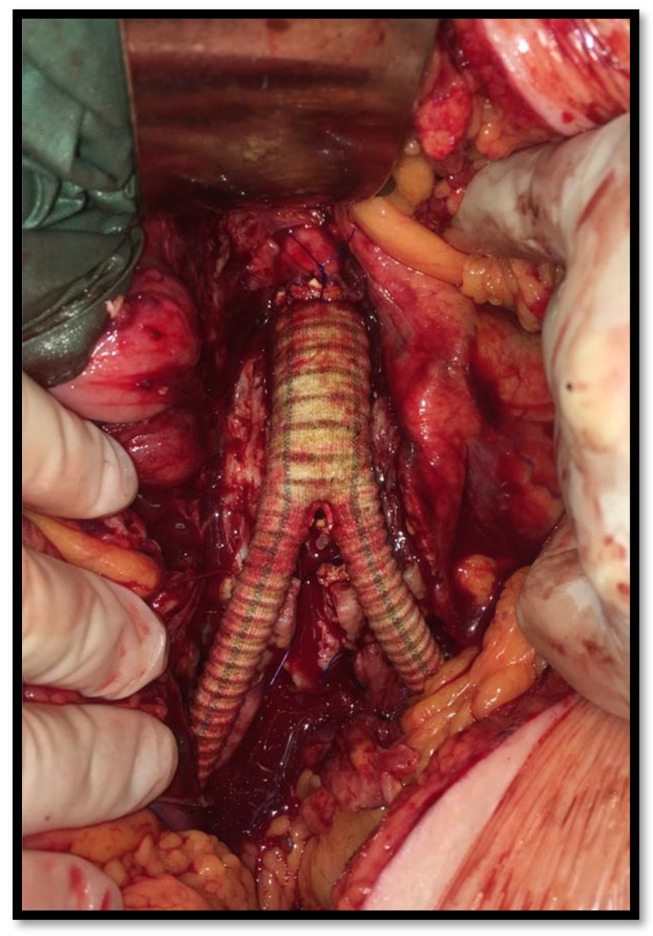
Silver-impregnated graft implanted after mycotic AAA resection.

**Figure 3 antibiotics-11-00386-f003:**
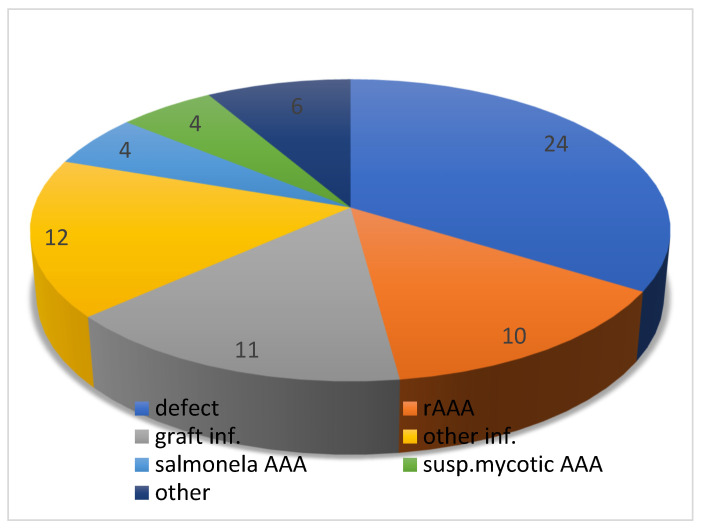
Indications for using silver-impregnated vascular grafts.

**Figure 4 antibiotics-11-00386-f004:**
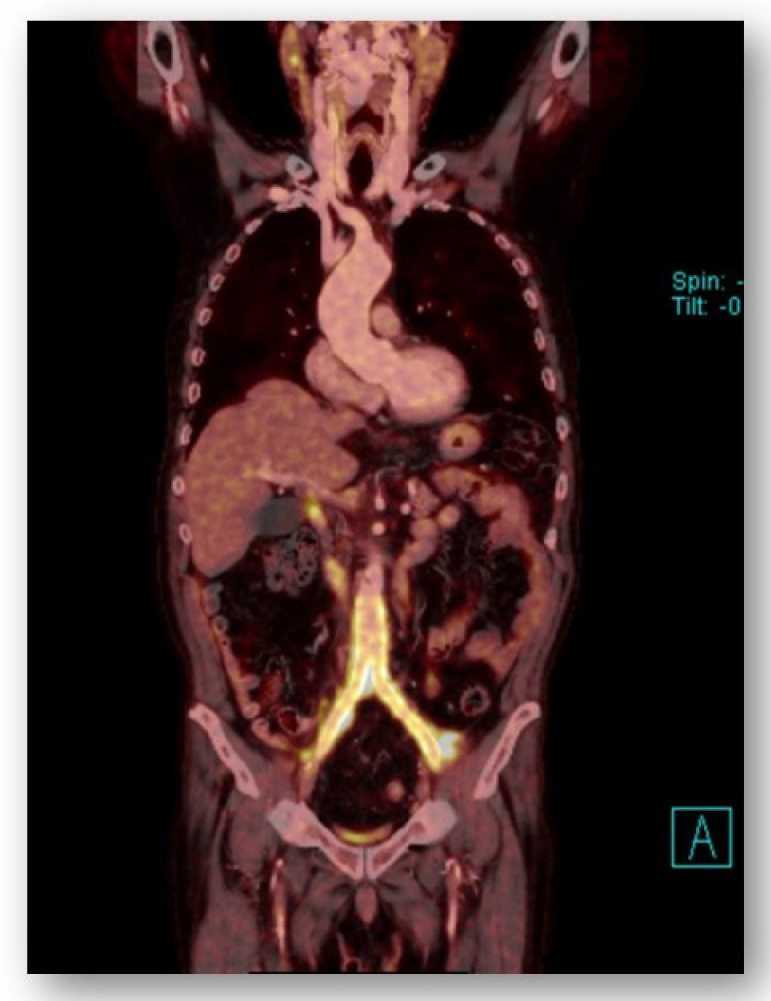
Infection of an aortobifemoral graft (PET/CT).

**Table 1 antibiotics-11-00386-t001:** Cohort of patients in the study.

Age (y)	Male	Female	Total
Age 40–60	11	6	17
Age 60–70	25	10	35
Age 70–80	14	2	16
Age above 80	3	0	3
Comorbidity			
DM	16	3	19
CHHF	18	5	23
CHRF	13	1	14
AH	39	11	50
ISS	5	0	5
ATB			
Cephazolin (prophylaxis)	25	10	35
Cefuroxime	5	2	7
Vancomycin	10	3	13
Ciprofloxacin	2	1	3
Combination	11	2	13
Type of Reconstruction			
Aortoiliac region	49	12	61
Peripheral region	4	6	10

AH—arterial hypertension, DM—diabetes mellitus, CHRF—chronic renal failure, ISS—immunosuppression situation, CHHF—chronic heart failure.

**Table 2 antibiotics-11-00386-t002:** Morbidity/mortality data in our cohort.

	Male	Female	Total (%)
30-Day Mortality	11	3	14 (19.7%)
Hemorrhagic shock	3	0	3
Multiorgan failure	6	2	8
Sepsis	1	1	2
Morbidity	23	7	30 (42.2%)
Respiratory failure	11	6	17
Multiorgan failure	10	3	13
Wound infection	15	3	18
Sepsis	6	0	10

**Table 3 antibiotics-11-00386-t003:** Data of patients with silver graft infections.

Sex/Age	Time to Reinfection	Diagnosis	Type of Reconstruction	Indication to SG	Infectious Agents	Solution	Result
Male/72	12 months	PET/CT positive cultures	Aortobifemoral graft	Rupture of mycotic AAA	*Salmonella enteritidis*	Re-operation, graft replacement with aortic allograft	OK
Female/68	6 months	PET/CT positive cultures	Femoropopliteal bypass	Lower limb gangrene	*Staphylococcus aureus*	Conservative treatment (ATB—vancomycin)	OK
Male/63	2 years	PET/CT positive cultures	Femoropopliteal bypass	Previous regular vascular graft infection	*Staphylococcus aureus*	Conservative treatment (ATB—vancomycin)	OK

## Data Availability

All data are available in University Hospital medical records.

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
