# Peer review of "Use of a Silver-Impregnated Vascular Graft: Single-Center Experience"

_antibiotics, 2022, doi:10.3390/antibiotics11030386_

Round 1

Reviewer 1 Report

1) Decimal point consistency check: for example 1,5-6%  versus 15.5%, as in Lines: 20,21, 23,25,46,83, etc.

2) Spell check: Line: 48, Ifections; 66, there for; 172, agens

3) Presentation: Total patients distributed across various treatment methods and categories make the numbers look diluted and their representation seem quite insignificant. Suggestion: Group some of the sub-categories together under one bigger cluster. Fewer groupings may highlight the differences better. Unless there are intentions to go into in-depth illustration of each category, listing all of them may cause distraction or confusion. Perhaps those numbers can be introduced in the passages instead since they are not used for further analysis.

4) "Silver graft is a bit higher in price than conventional ones, but lower compared to allograft or treatment of infectious complications". Perhaps  "optimum" may better describe it as a crucial factor instead of "minimal"(line 270)?

Author Response

Dear reviewer, 

thanks a lot for your comments, there are very helpful. My responses:

1) thanks, I have corrected it

2)  thanks, I have corrected it

3)  thanks for this suggestion, you are probably right. I was afraid abou the bias of small numbers. 

4) thanks, I have change it

Reviewer 2 Report

In the article: “Use of a silver-impregnated vascular graft: single-center experience”, Molacek et al. found out that silver grafts are a very suitable alternative for solving infectious or potentially infectious situations in vascular surgery. The aim of the article is very interesting, and the authors obtained valuable results, but some aspects must be improved before publication:

  1. Lines 48-50: “According to most authors, permanent and definitive treatment of an artificial vascular graft infection is impossible – you should add here at least 3 references to sustain this affirmation.
  2. The same for lines 50-52: “Therefore, graft explantation and replacement is almost always necessary. However, this is associated with the problem of replacement an infected graft. In the aortoiliac region venous graft cannot be used, with a few exceptions.”
  3. Sometimes there is a lot of textbook/Wikipedia-style writing. The information is enumerated in sentences that are not clearly connected; the text being confusing to readers. E.g., lines 86-87: Basic patients data summarized in Table 1.
  4. Please be more careful with some rules from the template: the title of the figures must be placed below, while the title of the table must be placed above.
  5. Starting with reference 13., you have used a different style.
  6. The manuscript needs a proofreading to correct the typographical and grammatical errors, because there are some punctuation and grammatical mistakes present throughout the manuscript.

Author Response

Dear reviewer, 

thanks a lot for your comments, there were very helpful. I tried to incorporate all of them into my revision. My responses:

1) I have quoted 3 important references proving my statement

2) the same

3) I have tried to correct a bit of manuscript style, I agree with you, I wanted the manuscript be clear and without redundant text.

4) thanks, I have corrected it 

5) thanks, corrected

6) thanks, I went throughout the manuscript and tried to correct it, also English speaking reviewer did corrections.

Reviewer 3 Report

The authors develop silver graft is a  very suitable alternative for solving infectious or potentially infectious situations in vascular surgery. Especially in urgent or acute cases, silver graft is often the only one option. In the reviewer's opinion, there are only a few comments and suggestions to be considered:

1)The introduction could be appropriately refined.

2) Whether the physical condition of the sample population has been taken into account

3)The sample number of data has credibility, and the sample parallelism, whether thereliability can be guaranteed

4) At present, compared with the existing methods, silver grafting technology innovation, new uniqueness, safety, biocompatibility should be emphasized.

Author Response

Dear reviewer, 

thanks a lot for your comments, there were very helpful and I incorporated them into my revision. My responses:

1) I have tried to correct the introduction

2) Yes, it is very important, I have added the information into manuscript

3) I think I can guarantee thereliability

4) I tried to mentioned it in discussion.

Reviewer 4 Report

The manuscript entitled "Use of a silver-impregnated vascular graft: single-center experience" describes the results of long-term studies and observations of patients who underwent surgery to install vascular grafts impregnated with silver. The authors cite long-term data on particularly risky cases, which is quite valuable scientific material and is of great interest. The article is written in a clear language and needs minor spelling corrections. However, there are a certain number of comments on the content of the article. First of all, there is a noticeably small number of references to literary sources, the authors' statements are often not supported by either their own data or links to other published articles. The article contains brief mentions of rather interesting unpublished data, which are not given in detail anywhere in the text. I would like to wish the authors, having such valuable material, to expand the article a little, adding their own data for comparison and increasing the literature review. It is also advisable to edit the content a little, especially fonts, figures and captions.

More detailed recommendations are given below.

1) Lines 35-39. You need to add links to the sources of this data.

2) Lines 15-25. Abstract is overloaded with methodological information. It can be specified briefly.

3) References to literary sources are not arranged in the text in order.

4) Lines 349-350. The bibliographic description is incomplete.

5) Lines 45-50. Data from 15-20 years ago are given. It is recommended to provide more recent data on this topic.

6) Lines 59-61. The results of these studies are quite clear. There is no obvious contradiction in them. Perhaps these data could be expanded by providing 5-10 links to studies, commenting on them and possibly dividing them into groups based on the results.

7) Lines 142-143. If the authors suppose that the data of 30-day mortality are a good result, it would be necessary to provide data with which it could be compared.

8) Line 146. It is necessary to remove the ellipsis.

9) Lines 145-148. The data with which the comparison takes place is not given.

10) Lines 168-169. It is necessary to add an comprehensive description of the PET/CT technique.

11) Figure 4. There are no captions to the figure.

12) Line 200. It would be necessary to provide confirmed mortality rates.

13) Lines 207-219. There are no references to literary sources on the basis of which statements are made.

14) Lines 217-219. The authors refer to their own unpublished data. At the same time, the data in the article is not provided. It would be desirable either to provide complete data by making a comparison, for example, using an analysis of variance, or to delete information about unpublished data.

15) Lines 215-216. A reference to the antibacterial activity of silver nanoparticles would be appropriate here, for example, to the article https://doi.org/10.3390/mi12121480 .

16) The article states that the transplants are impregnated with silver. It is not entirely clear whether silver salts or metallic silver are meant.

17) It is desirable to add a schematic image of the transplant to the article. The photo does not add clarity. Perhaps it would be worth illustrating the structure of its coating.

18) Table 3 is shifted to the left.

19) Figure 3. Captions are superimposed on the figure.

20) Figure 1 and Table 1 have captions at the top, the rest of the figures and tables have captions at the bottom. There is no uniformity in the design.

21) Table 1. The signature is located separately from the table.

22) Line 334. The font is changed.

Author Response

Dear reviewer, 

thank you very much for your comments, there were very helpful and i have tried to incorporate all of them into my revision of manuscript.

My responses:

1) thanks, I have added couple of important references

2) I totally agree with you, unfortunately other reviewer criticized me for short abstract without this crucial numbers

3) thanks, you are right, I am not sure if it is mandatory in this journal, I will check it and if yes, i will correct it

4) I am sorry, this reference is complete

5) thanks, you are right, I added more recent data

6) thanks, I added more data and did more comments about this issue

7) I have added date to compare our 30 day mortality

8) done

9) I have added references to compare the results

10) I am very sorry, it seems to me that PET CT is so well known, it is no necessary to use wide description of it´s technique but I make a short explanation

11) done

12) you are right but unfortunately I am not able to provide this confirmation

13) thanks, I have completed the reference comparing silver/atb graft

14) done

15) done

16) thanks, I tried to make clear in description - silver salts

17) you are right but all pictures descripting this technique are protected by trademark

18) yes, you are right, this table is to big, editor will fix it according his requirement

19) the same as 18

20) thanks, I have corrected it

21) corrected

22) corrected

Round 2

Reviewer 2 Report

The authors have taken into account all my recommendations. The article can be published in its current form.

Reviewer 3 Report

The authors evaluated the results of observational clinical study toward silver impregnated vascular graft in various indications. Overall, the manuscript provides guidance for solving infectious or potentially infectious conditions in vascular surgery. I would like to recommend the acceptance of the manuscript in Antibiotics.

Reviewer 4 Report

Thank you for your attention to my edits. Obviously, you fixed everything that was possible to fix. The essence of your research and its results are of scientific interest.